# Interleukin 6 SNP *rs1818879* Regulates Radiological and Inflammatory Activity in Multiple Sclerosis

**DOI:** 10.3390/genes13050897

**Published:** 2022-05-17

**Authors:** Antonio Bruno, Ettore Dolcetti, Federica Azzolini, Alessandro Moscatelli, Stefano Gambardella, Rosangela Ferese, Francesca Romana Rizzo, Luana Gilio, Ennio Iezzi, Giovanni Galifi, Angela Borrelli, Fabio Buttari, Roberto Furlan, Annamaria Finardi, Francesca De Vito, Alessandra Musella, Livia Guadalupi, Georgia Mandolesi, Diego Centonze, Mario Stampanoni Bassi

**Affiliations:** 1IRCSS Neuromed, 86077 Pozzilli, Italy; antonio.bruno91@yahoo.it (A.B.); ettoredolcetti@hotmail.it (E.D.); federica.azzolini@gmail.com (F.A.); stefano.gambardella@uniurb.it (S.G.); ferese.rosangela@gmail.com (R.F.); rizzo.francescaromana@gmail.com (F.R.R.); gilio.luana@gmail.com (L.G.); ennio.iezzi@neuromed.it (E.I.); giovalifi@gmail.com (G.G.); borrelliangela8790@gmail.com (A.B.); fabio.buttari@gmail.com (F.B.); f.devito.molbio@gmail.com (F.D.V.); m.stampanonibassi@gmail.com (M.S.B.); 2Department of Systems Medicine, Tor Vergata University, 00133 Rome, Italy; a.moscatelli@hsantalucia.it (A.M.); livia.guadalupi@gmail.com (L.G.); 3Laboratory of Neuromotor Physiology, IRCSS Fondazione Santa Lucia, 00179 Rome, Italy; 4Department of Biomolecular Sciences, University of Urbino “Carlo Bo”, 61029 Urbino, Italy; 5Clinical Neuroimmunology Unit, Institute of Experimental Neurology (INSpe), Division of Neuroscience, San Raffaele Scientific Institute, 20121 Milan, Italy; furlan.roberto@hsr.it (R.F.); finardi.annamaria@hsr.it (A.F.); 6Synaptic Immunopathology Lab, IRCCS San Raffaele Rome, 00163 Rome, Italy; alessandra.musella@uniroma5.it (A.M.); georgia.mandolesi@uniroma5.it (G.M.); 7Department of Human Sciences and Quality of Life Promotion, University of Rome San Raffaele, 00163 Rome, Italy

**Keywords:** multiple sclerosis, Interleukin 6, SNPs, CSF, neuroinflammation

## Abstract

*(1) Background:* The clinical course of multiple sclerosis (MS) is critically influenced by the expression of different pro-inflammatory and anti-inflammatory cytokines. Interleukin 6 (IL-6) represents a major inflammatory molecule previously associated with exacerbated disease activity in relapsing remitting MS (RR-MS); however, the role of single-nucleotide polymorphisms (SNPs) in the IL-6 gene has not been fully elucidated in MS. *(2) Methods:* We explored in a cohort of 171 RR-MS patients, at the time of diagnosis, the associations between four IL-6 SNPs (*rs1818879*, *rs1554606, rs1800797*, and *rs1474347*), CSF inflammation, and clinical presentation. *(3) Results:* Using principal component analysis and logistic regression analysis we identified an association between *rs1818879*, radiological activity, and a set of cytokines, including the IL-1*β*, IL-9, IL-10, and IL-13. No significant associations were found between other SNPs and clinical or inflammatory parameters. *(4) Conclusions:* The association between the *rs1818879* polymorphism and subclinical neuroinflammatory activity suggests that interindividual differences in the IL-6 gene might influence the immune activation profile in MS.

## 1. Introduction

Multiple sclerosis (MS) is a central nervous system (CNS) disease caused by an autoimmune chronic inflammation [1]. Inflammatory mediators play a key role in the pathophysiology of MS by promoting blood–brain barrier (BBB) damage, migration of innate and adaptive immune cells, and activation of neuroinflammatory cascade in the CNS [2]. Cytokines are a heterogeneous group of polypeptides that includes chemokines, lymphokines, interferons (IFNs), and growth factors involved in both pro-inflammatory and anti-inflammatory processes [2]. Cytokines, released by both peripheral and CNS resident immune cells, interact with a large number of receptors expressed by various cell types, including infiltrating lymphocytes and monocytes, microglia, astrocytes, endothelial cells, and neurons [2,3,4]. The complex functions played by these molecules under physiological and pathological conditions are regulated by key organizing principles [5]. These molecules constitute complex networks where the same cytokine can perform different activities depending on the inflammatory milieu, and the same effect can be mediated by several cooperating cytokines, according to a principle of redundancy [5]. For these reasons, it is extremely difficult to elucidate the role of each specific cytokine even though some molecules have been consistently associated with increased inflammation and a worse clinical course in MS. More specifically, interleukin (IL)-6 represents one of the most important pro-inflammatory cytokines in the pathophysiology of MS [2,6,7]. Preclinical studies in animal models of MS (i.e., experimental autoimmune encephalomyelitis, EAE), have shown that IL-6 deficient mice were fully resistant to the disease induction [8]. Similarly, blocking the IL-6 receptor (IL6R) led to a significant reduction of EAE symptoms [9]. Clinical studies in patients with relapsing remitting-MS (RR-MS) confirmed an association between higher levels of IL-6 in the cerebrospinal fluid (CSF) and a worse disease course characterized by an increased relapse rate and greater disability [6,7]. Notably, previous studies have shown that single nucleotide polymorphisms (SNPs) of the IL-6 gene can affect MS susceptibility [10,11]. These data suggest that interindividual IL-6 gene variability may influence the CSF inflammatory milieu and clinical presentation of MS. To explore this, we investigated whether four SNPs of the IL-6 gene (*rs1818879*, *rs1554606*, *rs1800797*, and *rs1474347*) are associated with different levels of CSF pro-inflammatory and anti-inflammatory molecules and clinical presentation in a group of RR-MS patients at the time of diagnosis.

## 2. Materials and Methods

### 2.1. MS Patients

In this study, we enrolled a group of 171 consecutive RR-MS patients at the time of diagnosis. We admitted patients to the neurological clinic of the Neuromed Research Institute in Pozzilli, Italy, between 2017 and 2019. The diagnosis of MS was made on the basis of clinical, laboratory, and MRI parameters. The Ethics Committee of the Neuromed Research Institute in Pozzilli, Italy approved the study according to the Declaration of Helsinki (cod. 06-17). All patients gave written informed consent to participate in the study. At the time of diagnosis, patients underwent a clinical evaluation, a brain and spine MRI, and a lumbar puncture. Clinical characteristics recorded were age, sex, an expanded disability status score (EDSS), the presence of clinical/radiological disease activity, and disease duration. Clinical activity was defined as the presence of a concomitant clinical relapse. Disease duration was defined as the interval elapsing between the first clinical episode compatible with MS and confirmed diagnosis.

### 2.2. IL-6 SNPs Analysis

Genotyping for IL-6 SNPs *rs1818879*, *rs1554606*, *rs1800797*, and *rs1474347* was performed in all enrolled patients. A blood sample (200 mL) was collected at the time of diagnosis. Genomic DNA was isolated from peripheral blood leukocytes according to standard procedures (QIAamp DNA Blood Mini Kit–QIAGEN, Hilden, Germany). IL-6 SNPs were analyzed with a TaqMan Validate SNP Genotyping Assay (Applied Biosystems, Foster City, CA, USA) using the ABI-Prism 7900HT Sequence Detection System (Applied Biosystems, Foster City, CA, USA) from 25 ng of genomic DNA in a final volume of 15 mL according to the manufacturer’s instructions.

### 2.3. CSF Collection and Analysis

In all RR-MS patients, CSF concentrations of inflammatory cytokines were analyzed. CSF was collected at the time of diagnosis, during hospitalization, and by lumbar puncture (LP). No corticosteroids were administered before LP. Disease modifying therapies were initiated after the confirmed diagnosis when indicated. CSF was stored at −80 °C and then analyzed using a Bio-Plex multiplex cytokine assay (Bio-Rad Laboratories, Hercules, CA, USA). CSF cytokines levels were determined according to a standard curve generated for the specific target and expressed as picograms/milliliter (pg/mL). Samples were analyzed in triplicate. The CSF cytokines analyzed included IL-1β, IL-2, IL-4, IL-5, IL-6, IL-7, IL-8, IL-9, IL-10, IL-12, IL-13, IL-15, IL-17, the tumor necrosis factor α (TNF-α), IFN-γ, the macrophage inflammatory protein 1α (MIP-1α), the macrophage inflammatory protein 1β (MIP-1β), the monocyte chemoattractant protein 1 (MCP-1), the granulocyte colony-stimulating factor (G-CSF), the granulocyte–monocyte colony stimulating factor (GM-CSF), the interleukin-1 receptor antagonist (IL-1ra), eotaxin, the fibroblast growing factor (FGF), the IFN-γ induced protein 10 (IP-10), the platelet-derived growth factor (PDGF), when regulated upon activation, normal T cells that are expressed and secreted (RANTES), and the vascular endothelial growth factor (VEGF).

### 2.4. MRI

All the patients underwent a 1.5T MRI scan of brain and spinal cord, which included the following sequences: dual-echo proton density, fluid-attenuated inversion recovery (FLAIR), T1-weighted spin-echo (SE), T2-weighted fast SE, and a contrast-enhanced T1-weighted SE before and after intravenous gadolinium (Gd) infusion (0.2 mL/kg). Radiological disease activity at the time of diagnosis was defined as the presence of Gd-enhancing (Gd+) lesions at the time of hospitalization in brain and spinal cord. 

### 2.5. Statistical Analysis

A Shapiro–Wilk test was used to evaluate normality distribution of continuous variables. Data were shown as mean (standard deviation, SD) or median (interquartile range, IQR). Categorical variables were presented as absolute (*n*) and relative frequency (%). A chi-square, or when necessary, a Fisher’s exact test, were employed to explore the associations between categorical variables. The difference in continuous variables between the IL-6 SNP groups was evaluated using a nonparametric Mann–Whitney test. A *p* value < 0.05 was considered statistically significant. When exploring the impact of SNPs on the CSF cytokine profile, we used a method based on dimensionality reduction (principal component regression) to first select a subset of cytokines for the second level analysis. Principal Component Analysis (PCA) was applied to the sample of the 27 CSF cytokines. Logistic regression was used to explore the association between principal components (PCs), each SNP, and to assess the association between SNPs and individual cytokines. All analyses were performed using IBM SPSS Statistics for Windows (IBM Corp., Armonk, NY, USA) and R (R Core Team).

## 3. Results

### 3.1. Clinical Characteristics in MS Patients

The clinical and demographic characteristics of RR-MS patients involved in the study are shown in Table 1. The first clinical event was characterized by: pyramidal symptoms (29.8%), visual symptoms (19.9%), brainstem symptoms (19.9%), cerebellar symptoms (8.2%), sphincteric symptoms (2.9%), and cognitive symptoms (1.2%). Missing data: 7 patients.

### 3.2. Analysis of IL-6 SNP

We assessed the frequencies of the alleles and genotypes for all IL-6 SNPs. In our cohort, the allele frequencies of the four SNPs were in the Hardy–Weinberg equilibrium considering the general Caucasian population (Gnomad database) (Table 2). To obtain two comparable groups, for each SNP, patients were divided into two groups, one homozygous for the major allele, and one carrying the minor allele in homozygosity or heterozygosity (Table 2). 

### 3.3. Association between CSF Inflammation and IL-6 SNPs

To explore whether individual genetic variability in the IL-6 gene could influence central inflammation in MS, we analyzed the possible association between IL-6 SNPs *rs1818879*, *rs1554606*, *rs1800797*, and *rs1474347* and the CSF cytokine profile. 

PCA, a dimension reduction technique which generates latent variables (PCs), was applied to our set of 27 cytokines [12]. The first six PCs explained 70.364% of the variance (Appendix A) and were retained for further analysis. In Figure 1 (Panel a and b), we show the association of specific cytokines with the first 4 PCs. We used logistic regression to assess the association between each SNP (*rs1818879*, *rs1554606*, *rs1800797*, and *rs1474347*) and the first six PCs. We found a significant association between *rs1818879* and PC1 (*β-coefficient* = 0.27; SE = 0.11; *p* = 0.018) (see also Appendix A). Conversely, we found no significant associations between the other SPNs explored.

As shown in Figure 1a, the following cytokines have high positive loading (cut-off value > 3) on PC1: IL-1β, IL-4, IL-5, IL-7, IL-9, IL-10, IL-13, G-CSF, PDGF, and VEGF. When analyzing the impact of *rs1818879* on the CSF levels of these cytokines, we found significantly higher levels of IL-1β (*p* = 0.0385); IL-9 (*p* = 0. 0231); IL-10 (*p* = 0. 0345); and IL-13 (*p* = 0.0319) in the A carrier. Conversely, the association between other cytokines was not significant (Table 3). 

### 3.4. rs1818879 Influences Radiological Activity in RRMS Patients

A significant association emerged between the SNP *rs1818879* and radiological activity at diagnosis (Table 4). In particular, the presence of the A allele was associated with a higher prevalence of gadolinium-enhancing lesions at the time of diagnosis (GG patients = 32.60%; GA/AA = 57.50; *p* = 0.001). We found no other significant differences between the two groups in the demographic and clinical characteristics examined.

## 4. Discussion

In the present study we investigated in a group of newly diagnosed RR-MS patients and the association between four SNPs of the IL-6 gene (*rs1818879*, *rs1554606*, *rs1800797*, and *rs1474347*) and a large set of CSF inflammatory molecules. PCA was applied to our set of 27 CSF cytokines to identify, in an unsupervised manner, specific components explaining the synergistic effect of different molecules [12]. We found a significant association between *rs1818879* and the first component (PC1) which represents the greatest source of variation, explaining 24.23% of the variance within our CSF cytokine set. PC1 reflects the combined effect of a large set of pro- and anti-inflammatory molecules including IL-1*β*, IL-4, IL-5, IL-7, IL-9, IL-10, IL-13, G-CSF, PDGF, and VEGF. In particular, the CSF levels of IL-1*β*, IL-9, IL-10, and IL-13 were significantly higher in A minor allele carriers of *rs1818879*. These data suggest that individual variability of IL-6 *rs1818879* may influence the CSF inflammatory milieu in RR-MS. In particular, A minor allele carriers may present with higher levels of central inflammation at the time of diagnosis.

When exploring the association between *rs1818879* and clinical characteristics, we found a higher prevalence of radiological disease activity among patients carrying the A minor allele. Conversely, no significant associations emerged with other clinical features, including EDSS and clinical activity. Altogether, these apparently contrasting findings may possibly suggest an increased susceptibility to new inflammatory subclinical episodes in these patients. 

In *rs1818879*, A minor allele carriers significantly increasing CSF levels of both pro-inflammatory and anti-inflammatory cytokines have been observed. Notably, IL-1β is a prototypical pro-inflammatory molecule involved in the migration of activated inflammatory cells into the CNS by altering BBB permeability [13]. This cytokine is produced by several immune cells including monocytes, macrophages, dendritic cells, neutrophils, T lymphocytes, and glial cells, in response to inflammatory signals [13]. Previous studies have clearly demonstrated the role of IL-1β in the pathogenesis of EAE and MS [14]. In particular, IL-1β detectability in the CSF of stable RR-MS patients has been associated with increased prospective disability and higher neurodegeneration [15]. The other cytokines associated with *rs1818879*, particularly IL-9, IL-10 and IL-13, have been classically linked to anti-inflammatory functions in MS [16,17,18,19]. In particular, IL-9 is secreted by T helper cells and regulates the balance between Th17 and T regulator (Treg) cells favoring the latter [20]. Similarly, IL-10 and IL-13 are pleiotropic, and immunoregulatory cytokines associated with T helper 2 and Treg cell responses and functions, promote immune homeostasis and anti-inflammatory responses [18,19]. A concurrent elevation of both pro- and anti-inflammatory cytokines may therefore suggest heterogeneous activation of the immune response in these patients. 

Previous studies have investigated the role of SNPs in the IL-6 gene in MS [10,11,21,22,23,24,25,26] and other autoimmune diseases (e.g., rheumatoid arthritis and erythematous systemic lupus [27,28]). Particularly for MS, the SNP *rs1800795*, located in the promoter region of the IL-6 gene, has been associated with MS risk and severity [11,21]. *rs1800795* has also been implicated in the development of optic neuritis risk [22,23], and in the modulation of flu-like symptoms in patients treated with interferon β1a [22]; however, the role of most IL-6 gene SNPs in MS is still unknown.

To the best of our knowledge, this is the first study demonstrating a direct effect of a SNP of the IL-6 gene on CSF cytokine milieu in RR-MS, and the first study investigating the role of *rs1818879* in MS. Previous studies have shown a possible a role of this polymorphism in different inflammatory conditions. In *rs1818879*, A minor allele carriers have a greater risk of developing inflammatory diseases such as chronic obstructive bronco pneumopathy (COPD) has been reported, which is associated with smoking [29]. In addition, a study showed a higher risk of developing major depressive disorder in patients with childhood maltreatment carrying the A minor allele of *rs1818879* [30].

Although our results indicate that *rs1818879* may significantly influence an immune response in MS, and may represent a possible marker associated with higher risk of neuroinflammation and disease activity, the lack of correlation between this SNP and IL-6 CSF concentrations represents an unexpected result. *rs1818879* is localized in the 3′ untranslated region (3′UTR) near the promotor of the IL-6 gene and the CCCTC-Binding factor (CTCF) binding site that can be involved in the modulation of gene expression [31]. As reported by the Genotype-Tissue Expression (GTEx) project, this SNP is localized in another gene placed in the opposite strand of IL-6 gene encoding for *AC073072*, a novel antisense long non-coding RNA, about which, little is known [31,32]. Although the mechanism remains unclear, localization in sites involved in the direct or indirect regulation of gene expression, suggests that *rs1818879* may be a functional polymorphism [31]. In this regard, one hypothesis could be that *rs1818879* is not involved directly in the synthesis of IL-6 but it could be able to indirectly influence the levels of other CSF cytokines; however, the idea that the lack of association with IL-6 levels could be due to statistical/technical limitations cannot be overlooked. Therefore, it is necessary to study a larger cohort of MS patients with homogeneous clinical characteristics, such as disease duration and activity, which have been previously associated with increased IL-6 expression [6,33]. Equally important, are studies with longer follow-ups, which are needed to clarify possible associations between A minor allele presence and clinical activity, as the effects of chronic increased levels of IL-1*β*, IL-9, IL-10, and IL-13, mediated by *rs1818879*, may influence the disease course in the long run. Other important limitations are represented by the lack of detailed MRI data, such as quantification of the T2 lesions’ load and volume.

In conclusion, the association between IL-6 *rs1818879* SNP and central inflammation suggests a role for this polymorphism in regulating disease activity in MS.

## Figures and Tables

**Figure 1 genes-13-00897-f001:**
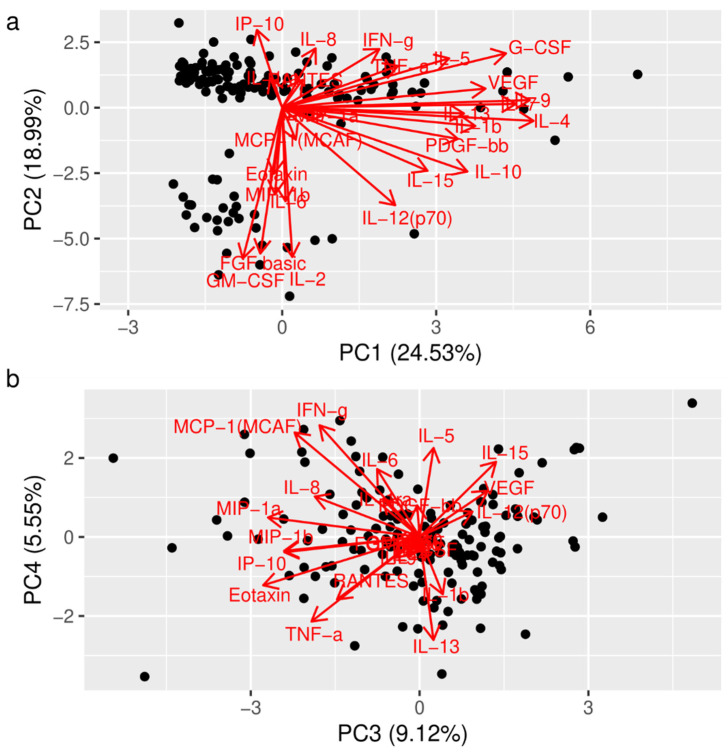
The association of specific cytokines with the first four PCs. Figure Legend: The biplot shows the orientation of the different cytokines with respect to the first and the second component (**a**), and the third and the fourth component (**b**), respectively. The biplots showing the orientation of the cytokines concerning the first four PCs. Abbreviations: PC (principal component); IL (interleukin); TNF (tumor necrosis factor); IFN (interferon); MIP (macrophage inflammatory protein); MCP (monocyte chemoattractant protein); G-CSF (granulocyte colony-stimulating factor); IL-1ra (interleukin-1 receptor antagonist); FGF (fibroblast growing factor); IP-10 (interferon γ induced protein 10); PDGF (platelet-derived growth factor); RANTES (regulated upon activation, normal T cell expressed and secreted); VEGF (vascular endothelial growth factor).

**Table 1 genes-13-00897-t001:** Demographic and clinical characteristics of RR-MS patients.

		MS Patients *n = 171*
**Sex, F**	N (%)	113 (66.10)
**Age, years**	Mean, (SD)	35.78 (12.27)
**Disease duration, months**	Median, (IQR)	5.10 (1.05–24.89)
**EDSS at diagnosis**	Median, (IQR)	2 (1–2.5)
**OCB presence, yes**	N (%)	132/166 (79.50)
**Radiological activity at diagnosis**	N (%)	74/166 (44.60)
**Clinical activity at diagnosis**	N (%)	68 (39.76)

Abbreviations: female (F), multiple sclerosis (MS), relapsing remitting (RR), expanded disability status scale (EDSS), interquartile range (IQR). Missing data: OCB (5 patients, 2.9%), Radiological activity (5 patients, 2.9%).

**Table 2 genes-13-00897-t002:** SNP distribution and allele frequency for the Hardy–Weinberg equilibrium of RR-MS patients calculated to single SNP and SNP groups for analysis.

SNP	SNP Distribution	Allele Frequency (%)	Chi-Square	Group Analysis (*n*)
*rs1818879*	GG (*n* = 88; 51.46%) AG (*n* = 79; 46.19%) AA (*n* = 4; 2.33%)	G = 75.44 A = 24.55	*p* = 0.917	GG (88) AG/AA (83)
*rs1554606*	GG (*n* = 78; 70.38%) TG (*n* = 46; 32.11%) TT (*n* = 11; 6.58%)	G = 74.81 T = 25.18	*p* = 0.842	GG (78) TG/TT (57)
*rs1800797*	GG (*n* = 86; 50.58%) AG (*n* = 74; 41.17%) AA (*n* = 10; 5.88%)	G = 72.35 A = 27.64	*p* = 0.886	GG (86) AG/AA (84)
*rs1474347*	AA (*n* = 84; 49.70%) CA (*n* = 75; 44.37%) CC (*n* = 10; 5.91%)	A = 71.81 C = 28.19	*p* = 0.870	AA (84) CA/CC (85)

Abbreviations: single nucleotide polymorphism (SNP). Missing data: *rs1554606* (4 patients, 2.34%), *rs1800797* (1 patient, 0.58%), *rs1474347* (2 patients, 1.17%).

**Table 3 genes-13-00897-t003:** Median (IQR) of cytokine levels with high loading in PC1 according to the SNP *rs1818879* group.

	GG	AG/AA	*p* Value	*β-Coefficient*	*SE*
**IL-1*β***	0.01 (0.01–0.05)	0.025 (0.00–0.07)	*p* = 0. 0385 *	7.00	3.38
**IL-4**	0.08 (0.01–0.15)	0.08 (0.00–0.22)	*p* = 0.104	1.25	0.770
**IL-5**	0.34 (0.00–2.16)	1.15 (0.00–3.34)	*p* = 0.0543	0.142	0.0740
**IL-7**	0.41 (0.00–0.92)	0.20 (0.00–1.41)	*p* = 0.0991	0.180	0.109
**IL-9**	1.86 (1.11–2.77)	2.36 (1.45–5.44)	*p* = 0.0231 *	0.105	0.0462
**IL-10**	1.78 (0.97–2.60)	2.11 (1.27–2.70)	*p* = 0.0345 *	0.278	0.132
**IL-13**	1.63 (1.04–3.32)	2.06 (1.11–4.53)	*p* = 0.0319 *	0.128	0.0597
**G-CSF**	15.29 (4.41–25.92)	16.51 (3.62–28.34)	*p* = 0.198	0.0131	0.0102
**PDGF**	0.00 (0.00–0.37)	0.00 (0.00–0.52)	*p* = 0.254	0.131	0.115
**VEGF**	4.03 (0.00–13.97)	5.79 (0.00–50.29)	*p* = 0.0715	0.00988	0.00548

Table Legend: Subjects carrying the homozygous major allele of SNP (GG), subjects carrying A minor allele (AG/AA) for SNP *rs18188792*. (*) denotes statistical significance (*p* < 0.05) using logistic regression analysis. Abbreviations: interquartile range (IQR); standard error (SE); IL (interleukin); G-CSF (granulocyte colony stimulating factor); PDGF (platelet-derived growth factor); VEGF (vascular endothelial growth factor). CSF analysis missing data: GG group (3 patients, 3.4%), AG/AA group (7 patients, 8.4%).

**Table 4 genes-13-00897-t004:** Demographic and clinical characteristics of RR-MS patients according to the SNP *rs1818879* group.

		GG *n = 88 (51.46%)*	AG/AA *n = 83 (48.53%)*	*p* Value
**Sex, F**	N (%)	57 (64.80)	56 (67.50)	*p* = 0.710
**Age, years**	Mean, (SD)	37.20 (12.38)	34.27 (12.04)	*p* = 0.111
**Disease duration, months**	Median (IQR)	6.66 (1.3–26.13)	3.1 (0.90–24.60)	*p* = 0.227
**EDSS**	Median (IQR)	2 (1–2.5)	2 (1–2.25)	*p* = 0.647
**OCB presence, yes**	N (%)	70/87 (80.50)	62/79 (78.50)	*p* = 0.752
**Radiological activity at diagnosis**	N (%)	28/86 (32.60)	46/80 (57.50)	*p* = 0.001 *
**Clinical activity at diagnosis**	N (%)	34 (38.63)	34 (40.96)	*p* = 0.707

Table legend: Subjects carrying homozygous major allele of SNP (GG), subjects carrying A minor allele (AG/AA) for SNP *rs18188792*. (*) denotes statistical significance (*p* < 0.05) using a nonparametric Mann–Whitney test for continuous variables and a chi-square test for categorial variables. Abbreviations: multiple sclerosis (MS), relapsing remitting (RR), expanded disability status scale (EDSS), interquartile range (IQR).

## Data Availability

Anonymized datasets are available upon reasonable request to the corresponding author.

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
