# Peer review of "Interleukin 6 SNP rs1818879 Regulates Radiological and Inflammatory Activity in Multiple Sclerosis"

_genes, 2022, doi:10.3390/genes13050897_

Round 1
Reviewer 1 Report
The authors of the manuscript entitled “ Interleukin 6 SNP rs1818879 regulates radiological and inflammatory activity in multiple sclerosis” aimed to evaluate association of four interleukin IL-6 SNPs (rs1818879, rs1554606, rs1800797 and rs1474347), with clinical and image diagnosis, and a relation with inflammatory profile in CSF of patients diagnosed with multiple sclerosis (MS). Authors observed a high association principal components (PCs) between genotype rs1818879 and inflammatry biomakers in CSF as well as MRI. Follow my considerations and concerns:
inserti “i” in ant (line 71)
Review line 164
Insert the information of the statistical analysis used in the legends of tables and figures.
Improve the discussion section about the the lack of correlation between genotype rs1818879 with clinical assessment, and only with imaging MRI.
At line 211, authors discussed a possibly role in BBB permeability, however authors did not evaluate that.
Author Response
Q1. inserti “i” in ant (line 71)
A: Thank you for the suggestion. We have now inserted “i” in ant (line 71)
Q2. Review line 164
A: Thank you. We have now checked and corrected previous line 164.
Q3. Insert the information of the statistical analysis used in the legends of tables and figures.
A: Thank you. We have now added information on the statistical analysis in the table and in the figure legends.
Q4. Improve the discussion section about the lack of correlation between genotype rs1818879 with clinical assessment, and only with imaging MRI.
A: Thank you. According to the Reviewer’s suggestion we have now improved the discussion (lines 290-295).
Q5. At line 211, authors discussed a possibly role in BBB permeability, however authors did not evaluate that.
A: Thank you, according to the Reviewer’s suggestion we have now modified the sentence at line 238-240
Reviewer 2 Report
The present study aims to explore in a population of patients with recent diagnosis of multiple sclerosis, the different IL6-SNPs present and their potential clinical impact.
It is an original, well-written study, highlighting a novel IL6-SNP (rs1818879) compared to previous studies (rs1800795) and which seems to be associated with increased radiological disease activity. However, the study did not find a correlation between this SNP and IL6 levels.
Nevertheless, I would have a few comments:
- Could the authors define what is meant by "clinical activity at diagnosis"? Did the authors include radiologically isolated syndrome in their study? If not, it is difficult for the reader to understand that less than half of the patients had clinical activity. Clarification is needed. Could the authors report the class of the first clinical event (NO, brainstem, myelitis...)
- Also, the proportion of patients with OCBs is unusually low in the study population. How do the authors interpret this data?
- Was the reported EDSS irreversible or close/linked to a relapse?
- Could the author add a table or supplementary data reporting the complete results of logistic regression between SNP and PC? And report the estimate or the odd ratio and the confidence interval of the correlation between rs818879 and PC1?
- Table 3: could the authors specify which multiple comparison correction technique was used?
- Table 4: T2 lesion load/volume or T2 lesion number should be reported and statistically tested between the two groups.
- The lack of correlation between IL6-SNP and IL6 levels should also be discussed regarding the technical/statistical limitations of the study.
Minor points
- Figure 1 is barely legible. The quality should be improved (dpi, size etc.) and could you please avoid overlap between the vectors and the text.
- Specify whether only the brain was assessed by MRI or whether the spinal cord was also explored. Also, whether the gadolinium-positive lesions reported were on brain MRI only.
Author Response
Q1. Could the authors define what is meant by "clinical activity at diagnosis"? Did the authors include radiologically isolated syndrome in their study? If not, it is difficult for the reader to understand that less than half of the patients had clinical activity. Clarification is needed. Could the authors report the class of the first clinical event (NO, brainstem, myelitis...)
A: Thank you. We have now better defined “clinical activity at diagnosis” in Materials and Methods (line 84). We also have better specified that in the present study were included patients who had prior clinical episodes compatible with MS diagnosis (lines 81-84). However, as the time elapsing between the last clinical episode and hospitalization was variable, only some patients still had clinical symptoms (see also Q3). We did not include patients with radiologically isolated syndrome. Clinical presentation of the first clinical event has been now added in the Results section (lines 137-140)
Q2. Also, the proportion of patients with OCBs is unusually low in the study population. How do the authors interpret this data?
A: We agree with the reviewer that the prevalence of OCBs in our MS cohort it is slightly lower than previously reported. The diagnosis of MS was supported by cardinal McDonald criteria of dissemination in space and time. It is likely that low prevalence of OCBs of our cohort may be due to low disease duration.
Q3. Was the reported EDSS irreversible or close/linked to a relapse?
A: As for Q1, in the methods section, we now have better reported that all clinical characteristics, including the EDSS, were scored at time of lumbar puncture. We have also better defined “clinical activity” (lines 81-84). It is worth to note that almost 40% of patients were evaluated at the time of clinical relapse. In the remaining patients, clinical relapse occurred longer before hospitalization and EDSS score did not reflect acute relapse but indicated permanent disability.
Q4. Could the author add a table or supplementary data reporting the complete results of logistic regression between SNP and PC? And report the estimate or the odd ratio and the confidence interval of the correlation between rs818879 and PC1?
A: Thank you for the suggestion. We have now added a supplementary table reporting correlation between rs818879 and all PCs (Supplementary Table 2). We have also reported the β-coefficient and SE between rs818879 and PC1 in Supplementary Table 2 and in the text (lines 170).
Q5. Table 3: could the authors specify which multiple comparison correction technique was used?
A: Thank you. We have now better specified in the Methods that principal component regression was used to increase the power of the analysis and to select a subset of cytokines for the second level analysis (lines 126-128). This method was used in replacement of the classical correction for multiple comparison (Gareth James, Daniela Witten, Trevor Hastie, Robert Tibshirani. An Introduction to Statistical Learning: with Applications in R. New York: Springer, 2013).
Q6. Table 4: T2 lesion load/volume or T2 lesion number should be reported and statistically tested between the two groups.
A: We thank you for this interesting suggestion. Although it may add interesting information to our research, these data were not collected during patient recruitment. MRI scans were strictly aimed at diagnosing the disease and establishing radiological activity. We added this limitation in the Discussion (lines 293-294).
Q7. - The lack of correlation between IL6-SNP and IL6 levels should also be discussed regarding the technical/statistical limitations of the study.
A: Thank you. According to the Reviewer’s suggestion we have now added this limitation in the Discussion (lines 285-289.).
Minor points
Q8. - Figure 1 is barely legible. The quality should be improved (dpi, size etc.) and could you please avoid overlap between the vectors and the text.
A: Thank you. We have now improved the quality of Figure 1 and tried to avoid overlap between text and vectors.
Q9. Specify whether only the brain was assessed by MRI or whether the spinal cord was also explored. Also, whether the gadolinium-positive lesions reported were on brain MRI only.
A: Thank you. We have specified that MRI with gadolinium included both brain and spinal cord assessment throughout the text.
